# Comparative Proteomic Analysis of Potato Roots from Resistant and Susceptible Cultivars to *Spongospora subterranea* Zoospore Root Attachment In Vitro

**DOI:** 10.3390/molecules27186024

**Published:** 2022-09-15

**Authors:** Xian Yu, Richard Wilson, Sadegh Balotf, Robert S. Tegg, Alieta Eyles, Calum R. Wilson

**Affiliations:** 1New Town Research Laboratories, Tasmanian Institute of Agriculture, University of Tasmania, New Town, TAS 7008, Australia; 2Central Science Laboratory, University of Tasmania, Hobart, TAS 7001, Australia; 3ARC Training Centre for Innovative Horticultural Products, Tasmanian Institute of Agriculture, University of Tasmania, Hobart, TAS 7001, Australia

**Keywords:** *Spongospora subterranea*, *Solanum tuberosum*, label-free proteomics, DIA, zoospore root attachment, host resistance

## Abstract

Potato (*Solanum tuberosum* L.) exhibits broad variations in cultivar resistance to tuber and root infections by the soilborne, obligate biotrophic pathogen *Spongospora subterranea*. Host resistance has been recognised as an important approach in potato disease management, whereas zoospore root attachment has been identified as an effective indicator for the host resistance to *Spongospora* root infection. However, the mechanism of host resistance to zoospore root attachment is currently not well understood. To identify the potential basis for host resistance to *S. subterranea* at the molecular level, twelve potato cultivars differing in host resistance to zoospore root attachment were used for comparative proteomic analysis. In total, 3723 proteins were quantified from root samples across the twelve cultivars using a data-independent acquisition mass spectrometry approach. Statistical analysis identified 454 proteins that were significantly more abundant in the resistant cultivars; 626 proteins were more abundant in the susceptible cultivars. In resistant cultivars, functional annotation of the proteomic data indicated that Gene Ontology terms related to the oxidative stress and metabolic processes were significantly over-represented. KEGG pathway analysis identified that the phenylpropanoid biosynthesis pathway was associated with the resistant cultivars, suggesting the potential role of lignin biosynthesis in the host resistance to *S. subterranea*. Several enzymes involved in pectin biosynthesis and remodelling, such as pectinesterase and pectin acetylesterase, were more abundant in the resistant cultivars. Further investigation of the potential role of root cell wall pectin revealed that the pectinase treatment of roots resulted in a significant reduction in zoospore root attachment in both resistant and susceptible cultivars. This study provides a comprehensive proteome-level overview of resistance to *S. subterranea* zoospore root attachment across twelve potato cultivars and has identified a potential role for cell wall pectin in regulating zoospore root attachment.

## 1. Introduction

The soilborne obligate biotrophic plant pathogen, *Spongospora subterranea* f. sp. *subterranea*, is responsible for root and tuber diseases that cause quality reduction and yield losses in potato production [1,2,3,4,5,6,7]. *S. subterranea* disease management is difficult and requires a range of approaches, including crop rotation, chemical application, and the selection of disease- or pathogen-free seed tubers [8,9,10,11,12]. However, the most efficient strategy to control *S. subterranea* diseases is arguably the planting of resistant cultivars [13]. Despite recent research into understanding the biochemical processes underlying *Spongospora*–potato interactions [7,14], the mechanism of resistance to *S. subterranea* tuber and root infections has not yet been elucidated.

Proteomics has been shown to be a powerful tool for the discovery of potential resistance mechanisms and protein biomarkers involved in the response of host plants to pathogen infection [15]. For example, quantitative proteomics was used to explore potato resistance to bacterial wilt caused by *Ralstonia solanacearum* [16], leaf late blight disease caused by *Phytophthora infestans* [17], and wart disease caused by *Synchytrium endobioticum* [18]. In addition, a recent study by Balotf, Wilson, Tegg, Nichols, and Wilson [14] compared the in planta transcriptome and proteome of *S. subterranea* invading susceptible and resistant potato cultivars. Their results suggested that the downregulation of enzyme activity and nucleic acid repair in the resistant cultivar could be related to resistance to *S. subterranea*.

Initial zoospore root attachment is one of the most critical phases of disease development in *S. subterranea* [7]. In our previous study [19], we reported the development of a novel in vitro bioassay that efficiently assessed potato cultivar resistance to *S. subterranea* root disease based on the efficiency of zoospore root attachment. We showed that reduced zoospore root attachment will likely manifest as less severe tuber and root infections [19]. During this critical stage of early infection, zoospores bind to the outside of the host roots and inject their contents into the root’s cell wall [7]. Successful attachment of zoospores on potato roots either leads to the development of a plasmodium, which subsequently forms a zoosporangium and can subsequently release further secondary zoospores [20], or to the formation of root galls and production of resting spores [21]. To date, however, the basis for host resistance to *S. subterranea* zoospore root attachment is not well understood at the molecular level. To address this knowledge gap, we used label-free proteomic analysis to compare the root tissues of twelve potato cultivars with various resistance to zoospore root attachment, leading to the identification of a range of candidate pathways and proteins that may influence the host resistance to zoospore root attachment.

## 2. Results

### 2.1. Root Attachment of Different Potato Cultivars Subjected to S. subterranea Infection

Potato cultivars with a range of host resistance to *S. subterranea* infection were selected and assessed for zoospore root attachment using an in vitro assay; the results are summarised in Figure 1. Significant differences (*p* < 0.001) were detected amongst the twelve cultivars for zoospore root attachment. The mean scores of zoospore root attachment of R1 to R6 ranged from 1 to 3, and S1 to S6 ranged from 9 to 13. This shows that R1 to R6 are much more resistant to the zoospore root attachment than S1 to S6, providing the basis for further analysis at the proteome level.

### 2.2. Overview of the Proteins in Potato Roots Identified by Label-Free Quantitative Proteomics

Using a DIA-MS approach, 3723 proteins were quantified across the 48 samples comprising four replicates of each of the twelve cultivars (provided in Appendix A). According to the statistical analysis results, 626 proteins were significantly less abundant in resistant cultivars, whereas 454 proteins were significantly more abundant in resistant cultivars (Figure 2a, and listed in full in Appendix A). Initially, PCA of the dataset comprising all proteins showed only partial separation of resistant and susceptible cultivar samples (Figure 2b). Although samples from the susceptible cultivars clustered quite tightly, those from the resistant cultivars were more dispersed and, in particular, root samples R1, R2, and R3 overlapped with the samples from susceptible cultivars (Figure 2b). Subsequent PCA of the protein subset identified significant differences between resistant and susceptible cultivars, showing stronger separation of the two groups, but nonetheless indicated greater variation overall in the resistant cultivars (Figure 2c).

### 2.3. Overall Functional Classification of Differentially Abundant Proteins

Gene Ontology (GO) analysis was used to categorise the sets of differentially abundant proteins (DAPs) into groups according to molecular function (MF), cellular component (CC), and biological process (BP) GO terms (Figure 3a,b). In total, 19 functional categories were captured by the set of proteins that were significantly more abundant in resistant cultivars, including several related to oxidative stress (e.g., BP “response to oxidative stress” and MF “peroxidase activity”) and metabolic processes (e.g., CC “mitochondrion”) (Figure 3a). In contrast, GO terms related to protein biosynthesis such as CC “cytosolic ribosome” and BP “protein folding,” and chloroplast functions (e.g., CC terms “chloroplast stroma” and “chloroplast envelope”) were associated with DAPs that were less abundant in the resistant cultivars (Figure 3b).

### 2.4. Overall Pathway Analysis of Differentially Abundant Proteins

To better understand how the metabolism of potato roots differed between resistant and susceptible cultivars in this study, KEGG-based analysis was used to categorise the DAPs into metabolic and genetic information pathways. The KEGG pathway enrichment analysis further revealed common or specific pathways in the sets of DAPs either more or less abundant in the root tissues of resistant cultivars (Figure 4). In total, five pathways were identified as significant among the proteins abundant in resistant cultivars, while 17 pathways were identified as significantly less abundant among the proteins in resistant cultivars. Accordingly, for proteins more abundant in resistant cultivars, most proteins were related to metabolic pathways (*n* = 78) including biosynthesis of secondary metabolites (*n* = 46) and phenylpropanoid biosynthesis (*n* = 22) (Figure 4). For the proteins less abundant in resistant cultivars (Figure 4), two pathways were related to genetic information processing (aminoacyl-tRNA biosynthesis (*n* = 11) and the proteasome (*n* = 9)), while the remaining 15 significant pathways were also classified as metabolic pathways (*n* = 131) including secondary metabolite biosynthesis (*n* = 90), antibiotic biosynthesis (*n* = 61), and carbon metabolism (*n* = 50).

### 2.5. Differentially Abundant Proteins of Root Cell Wall and Pathway Analysis

In total, 39 DAPs involved in cell wall composition and modification were identified. Notably, the vast majority of them (*n* = 37) were more abundant in resistant cultivars (Table 1). Pathway analysis of the cell wall related proteins that were more abundant in resistant cultivars (Figure 5a) identified a number of significant pathways such as glycosaminoglycan degradation (*n* = 3 proteins), biosynthesis of secondary metabolite (*n* = 7) and phenylpropanoid biosynthesis (*n* = 7). Gene Ontology analysis of the cell wall DAPs that were more or less abundant in resistant cultivars according to their major biological functions are summarised in Figure 5b. In total, 30 functional categories were captured by the set of proteins that were significantly increased, including several GO terms related to oxidative stress (e.g., BP “response to oxidative stress” and MF “peroxidase activity”) and cell wall functions (e.g., BP “cell wall organization”, “cell wall biogenesis”, “cell wall modification”, and CC “plant-type cell wall”, “cell wall”) (Figure 5b). In contrast, three functional categories (CC “plasmodesma” and MF “heme binding”, “metal ion binding”) were associated with DAPs that were less abundant in resistant cultivars (Figure 5b). Notably, four categories involved in cell wall pectin biosynthesis and remodelling were associated with proteins that were more abundant in resistant cultivars, including MF “pectin acetylesterase activity”, “pectinesterase inhibitor activity”, “pectinesterase activity”, and BP “pectin catabolic process”.

### 2.6. Effects of Pectinase Treatment of Potato Roots on Zoospore Root Attachment

The results from proteomic analysis indicated a potential role for cell wall pectin in the process of zoospore root attachment; therefore, we assessed the effect of pectinase treatment on zoospore attachment to one resistant (Gladiator) and one susceptible (Iwa) cultivar. Potato roots treated with pectinase exhibited a dose-dependent reduction in zoospore root attachment compared with the control in both susceptible and resistant cultivars (Figure 6). Significant reductions in zoospore root attachment on both resistant and susceptible potato cultivars were observed with pectinase concentrations of 1 and 2 mg/mL, with no zoospore root attachment observed following 3 mg/mL of pectinase solution.

## 3. Discussion

Root infection of potato by *Spongospora subterranea* is an under-explored area of research, despite the impact of infection on potato yield and subsequent tuber disease. Previously, we developed an in vitro bioassay for the rapid screening of potato resistance to zoospore root attachment [19], the precursor to root infection. Using this assay in the current study, we demonstrated a very clear difference in zoospore root attachment between the six resistant and six susceptible cultivars selected. Subsequently, we used label-free proteomics to analyse root tissue from this set of twelve cultivars and identified proteins that were significantly different between the groups of resistant and susceptible potato cultivars. The zoospore root attachment assay revealed significant reductions in zoospore attachment in all resistant cultivars, but also some variation between cultivars, which may account for the greater dispersion in proteomic data for the resistant cultivars (Figure 2).

Analysis of the proteomic profile of potato roots revealed that most of DAPs which were increased in resistant cultivars were assigned to GO terms related to oxidative stress and metabolic processes, including “response to oxidative stress”, “peroxidase activity” and “mitochondrion”. Peroxidases are well-known pathogenesis-related proteins that protect host tissue from pathogen attack by producing physical barriers through mediating undefined cell wall components [22]. They are reportedly involved in oxidative stress induced by pathogenic agents and the activation of defence-related activities in potatoes [23]. Similarly, peroxidase activity has been found to play a key role in defending plants against bacterial and fungal pathogens [24]. Peroxidases are also involved in phenol oxidation, IAA oxidation, lignification, plant defence, and plant cell elongation regulation [25,26,27,28,29]. Increases in peroxidase activity have been correlated with resistance in many species including rice, tomato, and wheat. In these plant hosts, peroxidases are involved in the polymerisation of proteins and lignin or suberin precursors into plant cell walls, which could inhibit zoospore attachment and penetration [30,31]. For proteins assigned to metabolic processes in resistant cultivars, they have important roles in the metabolism of carbohydrates, amino acids, nucleotides, and vitamins. These metabolic processes take place in organelles including the cytosol, chloroplast, mitochondria, and peroxisomes [32].

KEGG pathway analysis of the DAPs that were increased in resistant cultivars identified metabolic pathways such as the biosynthesis of secondary metabolites, phenylpropanoid biosynthesis, cyanoamino acid metabolism, and galactose metabolism. It has been shown that lignin biosynthesis, which is part of the phenylpropanoid metabolic process, contributes to resistance against pathogens in plants [33]. Li et al. [34] showed that the phenylpropanoid pathway was associated with resistance to potato wart disease. The plasmodiophorid soilborne pathogen *Plasmodiophora brassicae*, which causes clubroot disease, can involve drastic changes in the cell wall composition of host roots [35,36]. Several genes involved in phenylpropanoid metabolic process and cell wall synthesis were also upregulated in the transcriptome analysis of clubroot-infected *Brassica oleracea* [36]. Therefore, the establishment of mechanical barriers such as cell wall reinforcement of the host root seems to be a part of the mechanism behind plants’ resistance/tolerance mechanisms against *P. brassicae* [37,38]. In a recent study, Balotf et al. [39] showed that the phenylpropanoid metabolic process plays a critical role in the resistance of potato cultivars against root infection by *S. subterranea*. Their transcriptome analysis revealed upregulation of the phenylpropanoid metabolic process and lignin genes in the resistant cultivar, but not in the susceptible cultivar [39]. Our results from the proteomic analysis of twelve potato cultivars significantly expand on these previous findings and further suggest that lignin synthesis and cell wall thickening in the potato roots is a considerable obstacle for *S. subterranea*. We concluded that both constitutive and responsive gene/protein expression strategies are used by potato plants to increase resistance against *S. subterranea*.

Our proteome study showed that several enzymes involved in pectin biosynthesis and remodelling were identified as more abundant in resistant cultivars (Figure 5b). This included pectin acetylesterase which, in tobacco (*Nicotiana tabacum*), serves as a key structural regulator by changing the precise status of pectin acetylation to impact the remodelling and physiochemical characteristics of the cell wall’s polysaccharides [40]. Pectinesterase (a pectolytic enzyme that hydrolyses the ester linkages in pectin molecules; Maldonado and Strasser de Saad [41]) activity and inhibitor activity were also abundant in resistant cultivars, as was the pectin catabolic process pathways, resulting in the degradation of pectin (Choi et al., 2020). Pectin on plant root cell walls has been demonstrated to induce the rapid attachment of *Phytophthora cinnamomic* zoospores, implying that pectin-like materials on plant root surfaces may act as a recognition signal, resulting in zoospore root attachment [42,43,44]. Our current in vitro study revealed that potato roots pre-treated with pectinase exhibited significant reductions in zoospore root attachment, which further suggests an important role of potato root pectin in host resistance to zoospore root attachment. In this study, the effect of pectinase treatment on root morphology and plant growth was not analysed. However, it would be interesting to investigate the potential for the in vivo manipulation of cell wall pectin in modifying zoospore attachment and protection.

In summary, our findings in this study provide a better understanding of the constitutive basis of host resistance to zoospore root attachment among potato cultivars, representing two ends of the spectrum of root resistance to zoospore attachment. We have further identified several candidate pathways and proteins that have the potential to influence the cultivar resistance to zoospore root attachment process. Moreover, we have confirmed the biological importance of root pectin for zoospore root attachment. An important issue unresolved in this study is how any of these proteins respond to in situ plant–pathogen interactions, which should be addressed in future research. However, this study is the first to examine the differences across a range of potato cultivars with different levels of resistance to *S. subterranea* on a proteomic level. This represents an important set of data from which to start exploring functional aspects of host resistance to *Spongospora* tuber and root infections.

## 4. Materials and Methods

### 4.1. Plant Materials

Twelve potato cultivars with differential response to zoospore root attachment [19] were selected for detailed analysis: six resistant (R) cultivars (‘Gladiator’, ‘Granola’, ‘Toolangi Delight’, ‘Russet Burbank Ruen’, and ‘Tolaas’) and six susceptible (S) cultivars (‘Iwa’, ‘Nicola’, ‘10086’, ‘Shepody’, ‘Ida Rose’, ‘Kranz’, and ‘Russet Nugget’). Plants were maintained in tissue culture in liquid potato multiplication (LPM) medium, growing under a 16 h photoperiod, using white fluorescent lamps (65 µmol/m^2^/s) at 22 °C. The constitutes of LPM medium include MS salts, 4.43 g/L; sucrose, 30 g/L; casein hydrolysate, 0.5 g/L; ascorbic acid, 0.04 g/L; pH 5.8.

### 4.2. Spongospora subterranea Inoculum Preparation and Zoospore Germination

Sporosori inocula were obtained from powdery scab-infected potato tubers of cultivar ‘Kennebec’ harvested from a commercial crop grown on the northwest coast of Tasmania, Australia, in 2020. Tubers were washed with running tap water and air-dried in a cool place for one to two days. Powdery scab lesions were scrapped as fine as possible using a scalpel and sifted through a 600 µm sieve. The resting spore inoculum samples were kept at room temperature in the dark.

Zoospores were released by incubating 3 g sporosori inoculum in 10 mL of Hoagland’s solution in a McCartney bottle at 15 °C in the dark. Hoagland’s solution was prepared following Falloon’s standardised recipe [2]. The constituents of Hoagland’s solution were dissolved in deionized water, including KNO_3_, 253 mg/L; Ca(NO_3_)_2_·4H_2_O, 722 mg/L; KH_2_PO_4_, 2.3 mg/L; MgSO_4_·7H_2_O, 120 mg/L; NH_4_NO_3_, 40 mg/L; Fe-EDTA, 20 mg/L; H_3_BO_3_, 140 µg/L; KCl, 400 µg/L; MnSO_4_·H_2_O, 63 µg/L; ZnSO_4_·7H_2_O, 115 µg/L; CuSO_4_·5H_2_O, 50 µg/L; and Na_2_MoO_4_·2H_2_O, 22 µg/L. Zoospore numbers were determined by taking a 1 µL subsample and counting the total number of zoospores present by light microscopy at 200× magnification (DM 2500 LED, Leica Microsystem, Wetzlar, Germany).

### 4.3. Zoospore Root Attachment Assay

Confirmation of the relative resistance to zoospore root attachment of each cultivar was obtained by undertaking an in vitro zoospore root attachment assay. Following a two-week growth period in LPM, three primary roots (technical replicates) excised from each plant (biological replicates) of each cultivar or clone were washed in deionized water. A 10 mm section from the lower maturation region of each root was taken. The washed root segments were transferred into a treatment container (70 mm diameter) and evenly immersed in 60 mL of deionized distilled water (DDW) containing 1000 zoospores/mL. This zoospore treatment was incubated for 48 h at 15 °C in the dark, which has previously been shown to be optimal for zoospore root attachment [19]. The cultivars and variants were examined in batches of eight, and each batch contained two reference cultivars (‘Iwa’ and ‘Gladiator’). Five randomly chosen fields of view were used to count the number of zoospores attached to each root segment under 400× magnification. This evaluation of each specific cultivar was carried out using three independent biological replicates (three plants of each specific cultivar or clone), with each biological replicate consisting of three technical replicates (three roots from each plant).

Zoospore root attachment scores for each cultivar/clone in the screenings were normalised against the reference cultivars, ‘Gladiator’ and ‘Iwa’, with the first batch screening serving as a reference point (G1 + I1) to adjust for across-batch differences. Cultivar/clone scores were further linearly scaled according to the reference point correction coefficient (*η*_n_) in each batch [19].
$$\eta_{n}=\frac{Gn+In\;}{G1+I1}$$

Following checks of normality and homogeneity of variance, all data were subjected to analysis of variance (ANOVA) using IBM SPSS Statistics 27. Zoospore root attachment scores were analysed using a one-way ANOVA followed by a protected Fisher’s LSD test to determine statistically significant differences at the 5% level (*p* = 0.05).

### 4.4. Protein Extraction and Peptide Sample Preparation

Root proteins extracted from all twelve potato cultivars were then compared. Plants were grown in LPM medium for four weeks to provide sufficient root tissue, after which roots were excised for protein extraction. There were four independent biological replicates (plants) per cultivar. The total root tissue taken from each individual plant was washed with DDW and homogenised using a Fast Prep-24 bead beater (4000 rcf for 60 s) in PowerBead tubes with ceramic 2.8 mm beads (Qiagen, Hilden, Germany) in 200 µL of protein extraction buffer (50 mM Tris-HCl (pH 7.5), 100 mM NaCl, 5% glycerol, 10 mM DTT) and 20 µL protease inhibitor (one tablet of cOmplete Mini EDTA-free; Roche Diagnostics, North Ryde, NSW, Australia). The extract was centrifuged at 12,000 rcf for 8 min at 4 °C, the supernatant was collected and 6 volumes of cold acetone (−20 °C) was added, and the sample was mixed by shaking the tubes gently five times. The precipitated protein sample was collected by centrifugation at 6800 rcf for 5 min at 4 °C. The pellet was washed three times in chilled acetone before being dissolved in lysis buffer (6 M urea, 2 M thiourea).

The plant protein samples were quantified using the Qubit protein assay (Thermo Fisher Scientific, Waltham, MA, USA) and diluted to 0.5 mg/mL in lysis buffer (6 M urea, 2 M thiourea). Aliquots of 30 µg protein were sequentially reduced using 10 mM DTT overnight at 4 °C, alkylated with 50 mM iodoacetamide for 2 h at ambient temperature, and then digested with 1.2 µg proteomics-grade trypsin/LysC (Promega, Madison, WI, USA) as per the SP3 protocol [45]. The digests were acidified by the addition of trifluoroacetic acid to 0.1%, and then centrifuged at 21,000 rcf for 20 min to collect peptides. Peptides were then desalted using ZipTips (Merck, Darmstadt, Germany), as per the manufacturer’s instructions.

### 4.5. Sample Proteomic Analysis, Data Processing and Analysis

Peptide samples of approximately 1 µg were separated and analysed using an RSLCnano Ultimate 3000 and Q-Exactive HF mass spectrometer fitted with a nanospray flex ion source (Thermo Scientific, Waltham, MA, USA), essentially as described previously [46]. DIA-MS raw files were processed using Spectronaut software v14.7) (Biognosys AG, Schlieren, Switzerland) using the directDIA experimental analysis workflow. A spectral library was first generated by searching the DIA-MS data against the *Solanum tuberosum* UniProt reference proteome (UP000011115), comprising 53,106 entries, using the Pulsar search engine. This library, comprising 33,236 non-redundant peptide sequences and 4746 protein groups, was then used for the targeted re-extraction of DIA-MS2 spectra and relative protein quantitation between samples. With the exception of excluding single-hit proteins, default Spectronaut settings were used for protein quantitation and normalisation.

The Spectronaut protein group pivot report was imported into Perseus software for further processing. First, protein intensity values were log_2_-transformed, and proteins identified in fewer than 50% of the samples were filtered out, with the remaining missing values then replaced using Perseus default settings. Differentially abundant proteins were identified based on t-test comparisons of all replicates (*n* = 4) of the six resistant cultivars and six susceptible cultivars, with an FDR < 0.05 and s0 value of 0.1 used as the criteria to define significant proteins. Gene Ontology classification and enrichment analysis of significant proteins was provided by the UniProt database (www.uniprot.org) and DAVID bioinformatics resources 6.8 (https://david.ncifcrf.gov; accessed in November 2020), and the KEGG database (www.genome.jp/kegg/) was used for pathway analysis. Perseus software was used to generate principal component analysis (PCA) and volcano plots.

### 4.6. Pectinase Treatment

To provide further evidence of a possible role of root surface pectin, cultivars ‘Iwa’ (S) and ‘Gladiator’ (R) were assessed for the impact of pectinase treatment of roots on the capacity and efficiency of zoospore root binding. Tissue-cultured plants were cultured in liquid potato multiplication (LPM) medium, growing under a 16 h photoperiod, using white fluorescent lamps (65 µmol/m^2^/s) at 22 °C. The constitutes of LPM medium include MS Salts, 4.43 g/L; sucrose, 30 g/L; casein hydrolysate, 0.5 g/L; ascorbic acid, 0.04 g/L; pH 5.8. Pectinase solutions were made at four concentrations containing 0, 1, 2, and 3 mg pectinase (P4716; Sigma-Aldrich, Bayswater, Australia) in 1 mL of 50 mM sodium acetate buffer with pH 5.0, respectively. The enzyme activity of pectinase at 37 °C is 0.68 ± 0.020 µmolmin^−1^mL^−1^ [47]. Three primary roots (technical replicates) from each individual plants (biological replicates) of each cultivar were collected from propagated plantlets and rinsed thoroughly with DDW. This experiment was performed with three technical and biological replicates. A segment of the lower part of the root maturation region trimmed to a length of 20 mm was selected from each individual root [19]. The root segments comprising each biological replicate were added into one of three 1.5 mL Eppendorf tubes. Subsequently, 45 µL 50 mM sodium acetate buffer and 5 µL pectinase solution were added into all the tubes. Then, all three tubes were incubated at 37 °C for 0.5 h [48]. This experiment was repeated at four selected concentration levels (0, 1, 2, and 3 mg/mL) of pectinase solution. All treated root segments were then assessed for efficiency of zoospore root attachment by the in vitro zoospore root attachment assay.

## Figures and Tables

**Figure 1 molecules-27-06024-f001:**
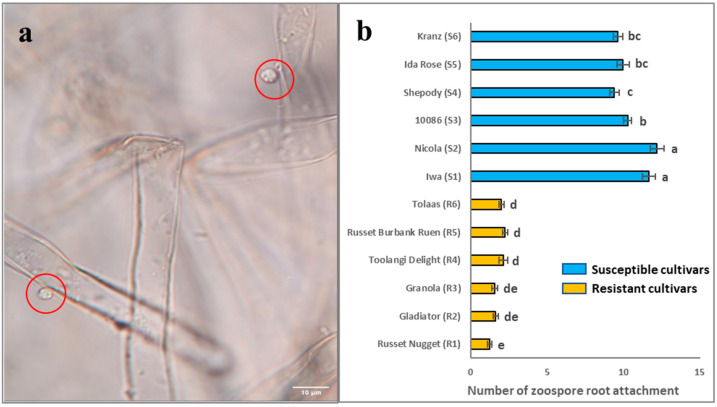
Zoospore root attachment of twelve potato cultivars. (**a**) Zoospore (in red circle) attached to potato root; (**b**) statistical analysis of zoospore root attachment severity scores of twelve potato cultivars ‘Iwa (S1)’, ‘Nicola (S2)’, ‘10086 (S3)’, ‘Shepody (S4)’, ‘Ida Rose (S5)’, ‘Kranz (S6)’, ‘Russet Nugget (R1)’, ‘Gladiator (R2)’, ‘Granola (R3)’, ‘Toolangi Delight (R4)’, ‘Russet Burbank Ruen (R5)’, and ‘Tolaas (R6)’ at 48 h after inoculation. Three independent biological replicates (from different plants) were assessed for each cultivar. Horizontal bars represent the standard error (*n* = 3). *p* < 0.001. LSD (0.05) = 0.73. The blue bars represent all the susceptible cultivars (S), and the yellow bars represent all the resistant cultivars (R) to zoospore root attachment. Bars that are labelled with different letters indicate values that are significantly different from each other.

**Figure 2 molecules-27-06024-f002:**
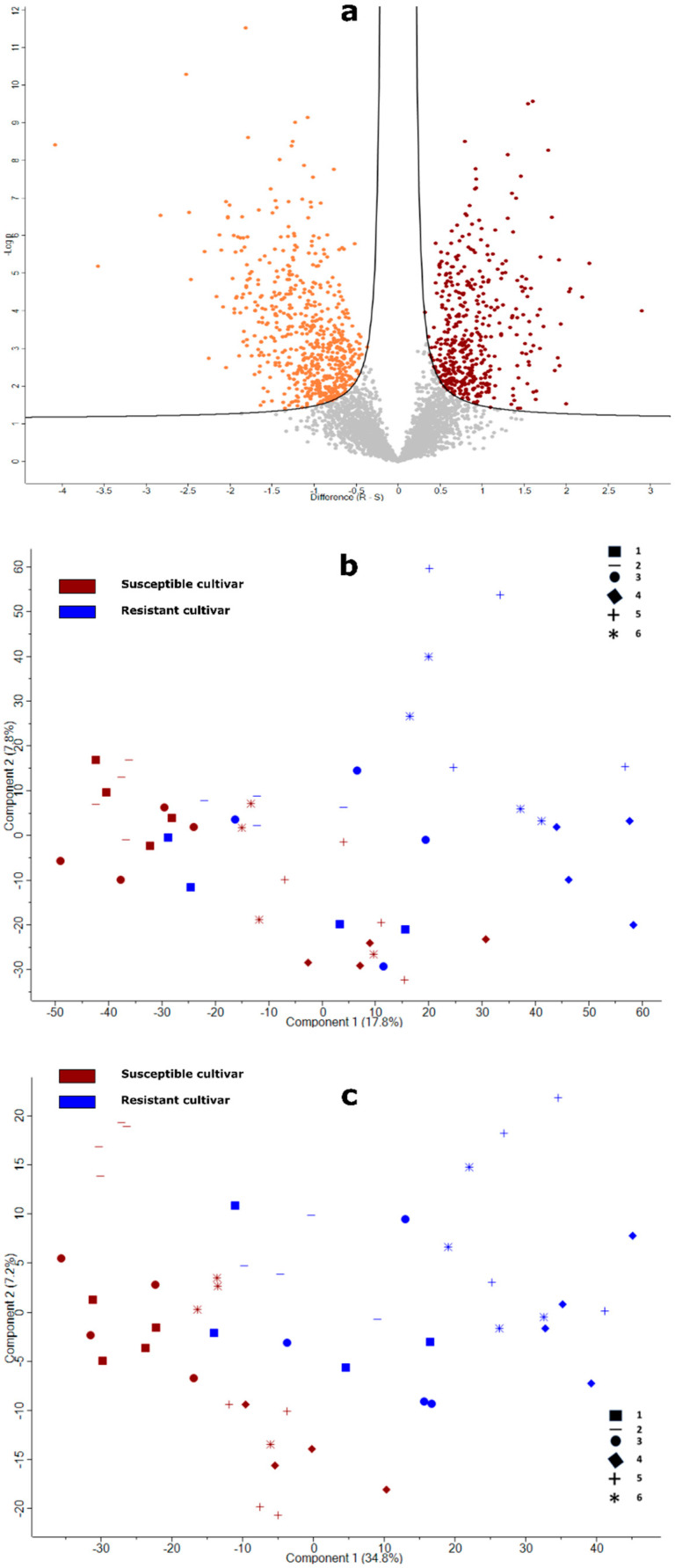
(**a**) Volcano plot displaying the results of *t*-test comparisons of susceptible and resistant potato cultivars. The two lines show the threshold (FDR < 0.05; s0 = 0.1) separating the proteins increased (dark red data points) and decreased in resistant cultivars (orange data points); (**b**) principal component analysis (PCA) of the dataset comprising all proteins quantified across the 12 potato cultivars; (**c**) PCA of the dataset restricted to the 1080 significant proteins between resistant and susceptible potato cultivars.

**Figure 3 molecules-27-06024-f003:**
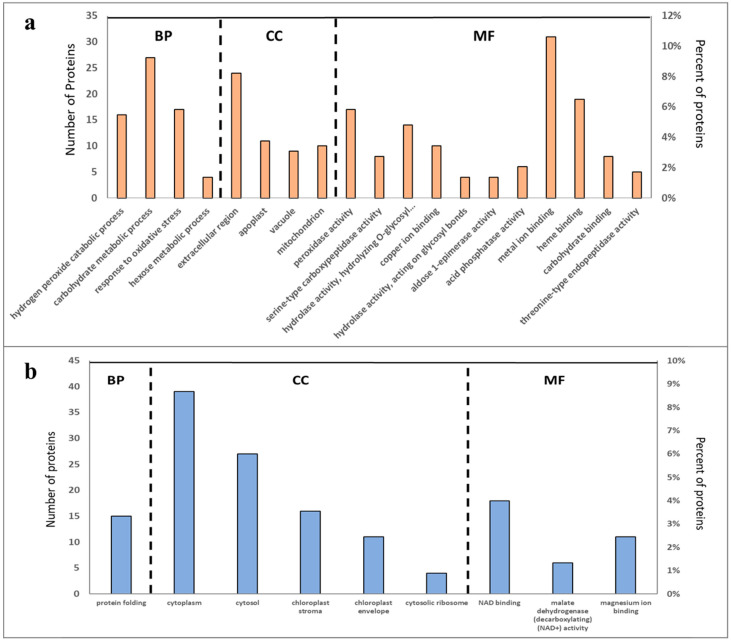
Classification of identified proteins of potato roots as (**a**) more abundant and (**b**) less abundant in resistant cultivars from the proteome of potato (Solanum tuberosum) into Gene Ontology (GO) categories, in terms of their involvement in biological process (BP, orange bar), cellular component (CC, green bar), and molecular function (MF, blue bar).

**Figure 4 molecules-27-06024-f004:**
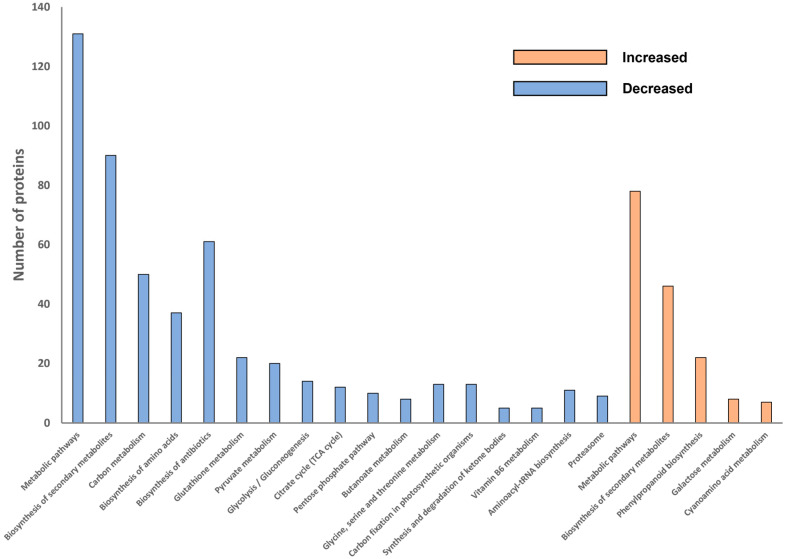
KEGG pathway classification and enrichment tests of proteins more or less abundant in resistant cultivars.

**Figure 5 molecules-27-06024-f005:**
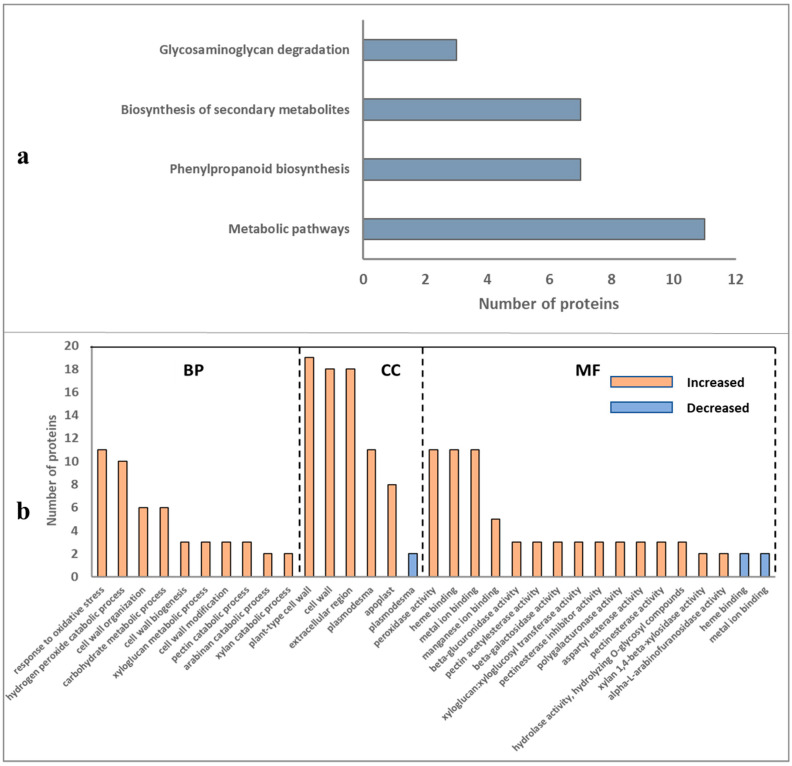
(**a**) Pathway analysis of cell wall related proteins that were increased in resistant potato (Solanum tuberosum) cultivars; (**b**) classification of root cell wall related proteins that were increased (orange bars) or decreased (blue bars) in resistant cultivars by Gene Ontology (GO) categories for biological process (BP), cellular component (CC), and molecular function (MF).

**Figure 6 molecules-27-06024-f006:**
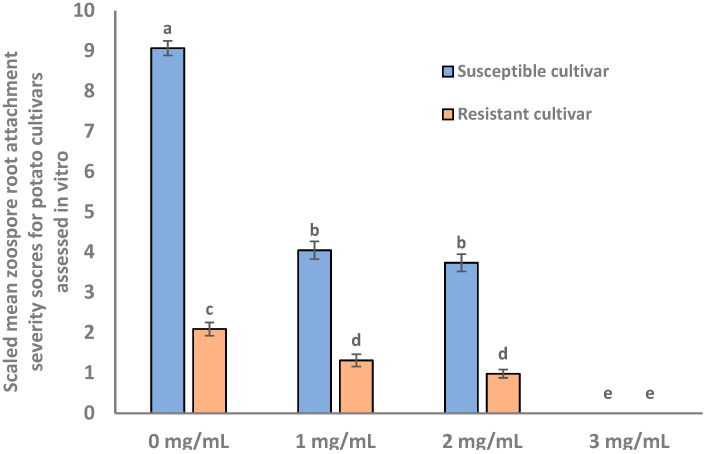
The concentration ranges of pectinase treated potato root segments to zoospore root attachment. The vertical bars represent standard error (*n* = 3). *p* (cultivars) < 0.001, *p* (concentration) < 0.001, *p* (cultivar × concentration) < 0.001. LSD (0.05) = 0.43. Bars that are labelled with different letters indicate values that are significantly different from each other.

**Table 1 molecules-27-06024-t001:** Differentially abundant proteins in cell wall. The fold change is on a log_2_ scale. Positive fold changes indicate increased abundance in resistant cultivars; negative fold changes indicate reduced abundance in resistant cultivars.

Accession	Protein Description	Fold Change	Adjusted *p*-Value
M1C976	Peroxidase	1.9	0.00
M1B051	Germin-like protein	1.7	0.00
M1BUZ0	Germin-like protein	1.5	0.00
M1A147	Beta-galactosidase	1.5	0.00
M1B041	Germin-like protein	1.1	0.00
M1BJ45	Pectinesterase	1.1	0.01
M1B6G3	Peroxidase	1.1	0.00
M1AQZ8	Xyloglucan	1.1	0.02
M1BFU7	Germin-like protein	1.1	0.00
M1D0Z2	Heparanase	1.0	0.00
M1BRR7	Pectin acetylesterase	1.0	0.00
M1AWV7	Polygalacturonase	1.0	0.00
M1C8D8	Pectin acetylesterase	1.0	0.00
M0ZQ51	Xyloglucan endotransglucosylase/hydrolase	0.9	0.03
M0ZJ69	Peroxidase	0.9	0.01
M1A385	Pectin acetylesterase	0.9	0.00
M1AZG9	Glycoside hydrolase family 28 protein	0.9	0.00
M1DTA0	Pectinesterase	0.9	0.02
M1BUZ2	Germin-like protein	0.9	0.00
M1AIV9	Pectinesterase	0.9	0.00
M1B6G2	Peroxidase	0.9	0.01
M1CV50	Expansin	0.9	0.01
M1CI69	Beta-galactosidase	0.8	0.00
M0ZGW4	Polygalacturonase	0.8	0.05
M1BTK5	Peroxidase	0.8	0.01
M1CE55	Peroxidase	0.8	0.01
M1A2Z2	Peroxidase	0.8	0.00
M1BYZ4	Peroxidase	0.8	0.01
M1AKA7	Periplasmic beta-glucosidase	0.8	0.01
M0ZJ70	Peroxidase	0.8	0.03
M1ARG0	Heparanase	0.7	0.02
M1BGD4	Xyloglucan	0.7	0.03
M0ZHI6	Beta-galactosidase	0.7	0.04
M1CAK9	Heparanase-2	0.6	0.00
M1CWU3	LEXYL2 protein	0.6	0.02
M1D155	Peroxidase	0.6	0.05
M1CCK1	Peroxidase	0.5	0.05
M1D1V1	Hemoglobin	−1.1	0.01
M1AY17	Peroxidase	−1.6	0.00

## Data Availability

Data are available via ProteomeXchange with identifier PXD022502 (Username: reviewer_pxd022502@ebi.ac.uk; Password: V8X3tODJ).

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
