# Peer review of "Comparative Proteomic Analysis of Potato Roots from Resistant and Susceptible Cultivars to *Spongospora subterranea* Zoospore Root Attachment In Vitro"

_molecules, 2022, doi:10.3390/molecules27186024_

Round 1
Reviewer 1 Report
Line 44. Spongospora subterranea provide the scientific name in abbreviated form.
Line 323, Please justify time of incubation. Why was selected incubation for 48 hours?
Line 342. Again , justify why the experiment was carried for four weeks.
Line 394 and 402. Please provided unity of enzymatic activity.
Line 214. First paragraph fits better in methodology
Line 214. Discussion. Please discuss the increased dispersion for resistant cultivars presented in volcano plots.
Discussion. Provide a summary figure with a model describing the main response of resistant cultivars, which might provide less susceptibility to zoospores attachment.
Author Response
Reviewer#1
Line 44. Spongospora subterranea provide the scientific name in abbreviated form.
Response: Accepted. The sentence has been revised.
Line 323, Please justify time of incubation. Why was selected incubation for 48 hours?
Response: The incubation time of 48 h was determined from a complementary study, which will be published soon titled: ‘Development and validation of a novel rapid in-vitro assay for resistance of potato cultivars to Spongospora subterranea zoospore root attachment’ (Plant Pathology, under revision). In that paper, we examined key factors sincluding incubation time, attachment location, and zoospore suspension concentration to develop a rapid and robust method for screening cultivar resistance to zoon spore attachment. We have revised the text to clarify the method as follows:
The washed root segments were transferred into a treatment container (70 mm diameter) and evenly immersed in 60 mL of deionized distilled water (DDW) containing 1000 zoospores/mL. This zoospore treatment was incubated for 48 hours at 15 C in the dark - which has previously been shown to be optimal for zoospore root attachment [19].
Line 342. Again, justify why the experiment was carried for four weeks.
Response: The sentence has been revised as ‘Plants were grown in LPM medium for four weeks to provide sufficient root tissue, after which roots were excised for protein extraction’.
Line 394 and 402. Please provided unity of enzymatic activity.
Response: Accepted. The sentence has been revised as ‘Enzyme activity of pectinase at 37oC is 0.68 µmolmin-1mL-1 [47].’
Line 214. First paragraph fits better in methodology
Response: In this instance we believe that this paragraph is required at the start of the Discussion as marks the importance of the in vitro zoospore root attachment assay used in this study and provides an important critical connection with the following proteome study.
Line 214. Discussion. Please discuss the increased dispersion for resistant cultivars presented in volcano plots.
Response: We thanks the reviewer for raising this interesting point and agree to include the following point in the Discussion:
The zoospore root attachment assay showed significant reduction in zoospore attachment in all resistant cultivars, but also some variation between cultivars, which may account for the greater dispersion in proteomic data for the resistant cultivars (Figure 2).
Discussion. Provide a summary figure with a model describing the main response of resistant cultivars, which might provide less susceptibility to zoospores attachment.
Response: Thanks for your helpful suggestion. However, we consider that further studies are required to validate the results of this study before a useful model can be developed
Reviewer 2 Report
This manuscript describes a comparative proteomic study of potato roots from cultivars with different resistance to Spongospora subterranean. An in vitro root attachment assay was used first to assess 12 potato cultivars and group them into a resistant group and a susceptible group. Then potato root samples from each cultivar were collected, and their protein abundance was quantified using DIA-MS approaches. Differentially abundant proteins (DAPs) identified from the resistant and susceptible groups were categorized based on molecular function, cellular component, and biological process GO terms. Finally, by focusing on pathways related to cell wall composition and modification, the authors found that many proteins up-regulated in the resistant group were associated with pectin biosynthesis and remodeling. Interestingly, a significant reduction in spore root attachment was observed in susceptible and resistant cultivars by treating potato roots with pectinase solution, implying an important role of pectin in host resistance to zoospore root attachment.
Overall, this paper identified several candidate pathways and proteins that might contribute to the cultivar resistance to the zoospore root attachment process. The studies described are well thought-out and well controlled. I recommend publication upon addressing the points detailed below.
Major points:
1. The effects of pectinase pretreatment on zoospore root attachment are intriguing and could be explored in further detail. For example, what are the effects of pectinase treatment on roots morphology and growth? Does the treatment provide a long-term zoospore attachment protection for potato roots? Does the pectinase have any effect on roots already infected by zoospores?
2. Besides pectinase, a few other enzymes involved in cell wall pectin remodeling have been shown to be up-regulated in the resistant group, such as pectin acetylesterase. It would be interesting to conduct a similar pre-treatment assay and look for if they have any effect on root zoospore attachment.
Minor points:
1. A typographical error in line 72. ‘Address’ was misspelled as ‘redress’.
2. Figure 3, Figure 4, and Figure 5 serve very similar purposes but were presented in various styles. These figures can be made more concise, i.e., adjusted as subfigures of 1 or 2 main figures. Meanwhile, the styles of these bar graphs should be adjusted to be more consistent.
Author Response
Major points:
- The effects of pectinase pretreatment on zoospore root attachment are intriguing and could be explored in further detail. For example, what are the effects of pectinase treatment on roots morphology and growth? Does the treatment provide a long-term zoospore attachment protection for potato roots? Does the pectinase have any effect on roots already infected by zoospores?
Response: The pectinase pretreatment was conducted to remove root surface pectin in an in vitro system using detached root segments. The primary purpose was to validate the results of the proteomic analysis and further investigate the potential role for cell wall pectin in resistance to zoospore root attachment. As such, the effect of this treatment on root morphology and growth in vivo was not analyzed as this was not the aim of this experiment. However, we agree with the reviewer that it would be interesting to follow up with in vivo experiments to further investigate the potential for manipulation of cell wall pectin in modifying zoospore attachment and protection.
- Besides pectinase, a few other enzymes involved in cell wall pectin remodeling have been shown to be up-regulated in the resistant group, such as pectin acetylesterase, It would be interesting to conduct a similar pre-treatment assay and look for if they have any effect on root zoospore attachment.
Response: Again, thanks for your excellent advice. We heartily agree with your comment and believe other enzymes may be involved in this process. However, considering this is the very first research on this particular topic, the knowledge presented here encourages further investigation of potential enzymes involved in cell wall pectin remodelling for cultivar resistance to zoospore root attachment.
Minor points:
- A typographical error in line 72. ‘Address’ was misspelled as ‘redress’.
Response: accepted and corrected accordingly.
- Figure 3, Figure 4, and Figure 5 serve very similar purposes but were presented in various styles. These figures can be made more concise, i.e., adjusted as subfigures of 1 or 2 main figures. Meanwhile, the styles of these bar graphs should be adjusted to be more consistent.
Response: Thanks for your suggestion. Accepted. The styles of Figure 3, Figure 4, and Figure 5 have been revised to be more consistent now. We believe that the current arrangement of these Figures, such that they are interspersed with the relevant parts of the text, contributes to the readability of the manuscript.